# Systematic Review of Incidence Studies of Pneumonia in Persons with Spinal Cord Injury

**DOI:** 10.3390/jcm11010211

**Published:** 2021-12-31

**Authors:** Anja Maria Raab, Gabi Mueller, Simone Elsig, Simon C. Gandevia, Marcel Zwahlen, Maria T. E. Hopman, Roger Hilfiker

**Affiliations:** 1Clinical Trial Unit, Swiss Paraplegic Centre, 6207 Nottwil, Switzerland; gabi.mueller@paraplegia.ch; 2Department of Health Professions, Bern University of Applied Sciences, 3008 Bern, Switzerland; 3School of Health Sciences Valais, Physiotherapy, University of Applied Sciences and Arts Western Switzer Land Valais, 1950 Sion, Switzerland; simone.elsig@hevs.ch (S.E.); roger.hilfiker@hevs.ch (R.H.); 4Neuroscience Research Australia, Randwick 2031, Australia; s.gandevia@neura.edu.au; 5School of Medical Sciences, University of New South Wales, Sydney 2052, Australia; 6Institute of Social and Preventive Medicine, University of Bern, 3012 Bern, Switzerland; marcel.zwahlen@ispm.unibe.ch; 7Department of Physiology, Radboud University Nijmegen, 6525 Nijmegen, The Netherlands; maria.hopman@radboudumc.nl

**Keywords:** pneumonia, spinal cord injury, systematic review, incidence

## Abstract

Pneumonia continues to complicate the course of spinal cord injury (SCI). Currently, clinicians and policy-makers are faced with only limited numbers of pneumonia incidence in the literature. A systematic review of the literature was undertaken to provide an objective synthesis of the evidence about the incidence of pneumonia in persons with SCI. Incidence was calculated per 100 person-days, and meta-regression was used to evaluate the influence of the clinical setting, the level of injury, the use of mechanical ventilation, the presence of tracheostomy, and dysphagia. For the meta-regression we included 19 studies. The incidence ranged from 0.03 to 7.21 patients with pneumonia per 100 days. The main finding of this review is that we found large heterogeneity in the reporting of the incidence, and we therefore should be cautious with interpreting the results. In the multivariable meta-regression, the incidence rate ratios showed very wide confidence intervals, which does not allow a clear conclusion concerning the risk of pneumonia in the different stages after a SCI. Large longitudinal studies with a standardized reporting on risk factors, pneumonia, and detailed time under observation are needed. Nevertheless, this review showed that pneumonia is still a clinically relevant complication and pneumonia prevention should focus on the ICU setting and patients with complete tetraplegia.

## 1. Introduction

Pneumonia continues to complicate the course of spinal cord injury (SCI). Currently, clinicians and policy-makers are faced with only limited numbers of pneumonia incidence in the literature. Respiratory complications are one of the main comorbidities after SCI, especially among persons with cervical and high thoracic injury [1]. The underlying problem is paralysis of the respiratory muscles, which leads to poor mobilization of secretion, bacterial accumulation in the secretion, and resultant respiratory infections [1,2,3]. The higher the level of SCI, the greater is the risk of respiratory complications [2]. About 30% of all deaths after an SCI are due to respiratory causes, with pneumonia as the most common respiratory cause [4].

Pneumonia is defined as inflammation of the lung tissue and is usually caused by infection [5,6]. The United States Centers for Disease Control and Prevention provide an overview of causes of pneumonia [7]. Pneumonia can be caused by viruses, bacteria, and fungi. Common causes of viral pneumonia are influenza viruses, respiratory syncytial virus, and SARS-CoV-2. A common cause of bacterial pneumonia is Streptococcus [8,9]. However, clinicians are not always able to find out which pathogen caused pneumonia. Generally, the bacteria and viruses that most commonly cause pneumonia in the community are different from those in healthcare settings [7]. Diagnosis can be made by radiographic signs of parenchymal disease [6] or clinical signs such as fever, inflammatory markers or purulent tracheobronchial secretions.

Pneumonia profoundly impacts the length of hospital stay and the neurological outcome in persons with SCI [10]. Many persons with SCI, who survive acute hospitalization, die within 6.2 years after discharge [11], mainly as a result of cardiovascular (13–37%) and pulmonary diseases (9–30%) [11,12,13]. In patients with community-acquired pneumonia, the case fatality rate for pneumonia (7.9% within 60 days) is greater in persons with SCI compared to the general population, and hospitalizations are more frequent with increasing age, tetraplegia, and the occurrence of comorbidities [14]. Male gender, motor complete injury, presence of chest trauma and the timing of intubation are key predictors for pneumonia in SCI [10].

The risk of pneumonia is the sum of different factors such as level of injury, the clinical setting, the use of mechanical ventilation, the presence of tracheostomy, and dysphagia. The higher the level of spinal-cord damage, the more severe are the respiratory impairment [15] and the risk of pneumonia. Respiratory dysfunction in SCI can be considered in 2 phases: (1) the initial phase immediately following the injury and the year thereafter, and (2) the chronic phase during the rest of the life of the affected individual [16]. Early after an injury, a reduction in lung compliance occurs, with reduced lung volumes and changes in the mechanical properties of the lung [16]. Brown et al. described an improvement of respiratory function with time depending on the level and completeness of the injury, the extent of the spontaneous recovery, and other factors [16]. Thus, the time post injury and the setting play an important role in the development of pneumonia. Patients affected by pneumonia can be admitted to ICUs independently by the setting where the infection has been acquired [17]. However, frequently pneumonia can develop in patients already in an ICU, especially in those requiring mechanical ventilation [17].

In persons with severe paralysis, the risk of dysphagia is increased, particularly in the first weeks after injury [18]. Mechanical ventilation and prolonged tracheostomy further increase the risk of pneumonia [19,20]. Shem et al. reported that 75% of spinal cord injured patients with dysphagia developed pneumonia compared to 29% without dysphagia [19]. Martin et al. showed that pneumonia is significantly associated with the need for a tracheostomy in 67% of patients with SCI [21].

A formal incidence of pneumonia in persons with SCI is still missing and, to our knowledge, this is the first systematic review of the incidence of pneumonia in the SCI population. Incidence of a disease indicates the number of new cases within a time period in a population. Systematic reviews of prevalence and incidence data are becoming increasingly important as decision-makers realize their usefulness in informing policy and practice [22]. Accurate estimates of the true incidence of pneumonia are also of value in improving the understanding and awareness in an SCI population and in planning diagnostic and intervention services. We hypothesized that the risk of pneumonia may be influenced by various factors such as the clinical setting, the level of injury, the use of mechanical ventilation, the presence of a tracheostomy, and the presence of dysphagia. Therefore, a systematic review of the literature was undertaken to provide an objective synthesis of the evidence about the incidence of pneumonia in persons with SCI using five covariates: clinical setting, the level of injury, use of mechanical ventilation, presence of tracheostomy, and dysphagia.

## 2. Materials and Methods

We conducted a systematic review and meta-regression of incidence studies of the incidence of pneumonia in SCI. The review was guided by the recommendations provided for Meta-analysis of Observational Studies in Epidemiology (MOOSE) and guidelines for undertaking systematic reviews of incidence and prevalence studies [23,24]. The review was registered with the International Prospective Register of Systematic Reviews (PROSPERO 2019 CRD42019129048).

### 2.1. Types of Studies

We considered all types of studies for inclusion, except case studies.

### 2.2. Literature Searches

A variety of sources were used to find relevant publications, including PubMed, EMBASE, MEDLINE (Ovid) and the Cochrane Central Register of Controlled Trials (CENTRAL) databases. Search terms and the combination of exploded Medical Subject Heading/Emtree terms using “or” and “and” per database specification are provided in Appendix A. One review author (AMR) designed this search strategy in collaboration with an experienced health librarian. A systematic and comprehensive search was scheduled on 20 March 2019 with a final update on 12 May 2020. We also searched the reference lists of relevant papers and literature reviews by hand. We contacted study authors to acquire information that was not included in their articles. Additionally, we contacted experts in the field to find all publications that matched our inclusion criteria. Initially we applied no date, language or publication restrictions. The search strategy for the six databases is provided in Appendix A. However, three papers in Chinese, Danish, and Icelandic could not be translated and they were not included in the analysis. No other studies required translation into English. Papers with low incidence may be under-represented in our final list of publications.

#### Eligibility Criteria

We included studies involving male and female patients with a primary diagnosis of traumatic or non-traumatic SCI, American Spinal Injury Association Impairment Scale (AIS) A–D, right and left motor level between C1-L5, both acute and chronic. The participants were 18 years of age and over. In those studies with missing information about the AIS grade, we contacted the authors of the study. For those who gave no answer we used the following definition: “complete” SCI indicates AIS A in which no motor or sensory function is preserved, and “incomplete” SCI indicates an AIS B,C,D in which sensory but no or partial motor function is preserved [25]. In the included studies, only the term “pneumonia” was used, and the causes and types of pneumonia were not specifically defined. Therefore, analysis among different types of pneumonia was not possible.

Studies on patients with progressive neurological diseases such as multiple sclerosis, poliomyelitis or amyotrophic lateral sclerosis were excluded as well as studies on patients with mental disorders, patients taking bronchodilators or any other medication that influences respiration at the time of assessment. We also did not include studies that investigated pneumonia caused by the recently discovered coronavirus with the outbreak in China in December 2019 and the ensuing pandemic.

Full-text, peer-reviewed studies were required to report data on incidence of pneumonia in persons with SCI. Studies could be conducted in the hospital or community setting.

### 2.3. Study Records

The search results were collated in an EndNote X8 database (Clarivate Analytics, Philadelphia, PA, USA). Duplicates were removed before search results were analyzed. Two review authors (A.M.R., S.E.) independently assessed the titles and abstracts to identify potentially relevant articles by using the Covidence systematic review software (Veritas Health Innovation, Melbourne, Australia). After initial screening, the two review authors independently assessed the full texts of the retrieved articles for compliance with the eligibility criteria. Disagreement was resolved by discussion. In cases when no decision could be made by consensus, a third author (G.M.) was consulted for discussion until agreement could be reached. A PRISMA flow chart of the study selection procedure was created (Figure 1).

### 2.4. Data Extraction

Methods and population characteristics reported across studies were selected for data extraction. One review author (A.M.R.) extracted and coded data from included studies by using a predefined form. A second review author (R.H.) checked the extracted data. The second review author consulted the first one in cases where there was disagreement to find a consensus. The following items were extracted: year of publication, country, design of study, sample size, clinical setting, age, sex, AIS grade, level of injury, length of hospital stay, use of mechanical ventilation, length of observation, and the incidence of pneumonia.

### 2.5. Data Processing

We calculated incidence rates per 100 person-days based on the number of events divided by the total exposure time in days. Because most of the studies only reported overall follow-up time and not the exact time at risk to develop pneumonia, we estimated the time at risk by taking the mean follow-up time in days for patients without pneumonia and by taking half of the mean follow-up time in days for the patients with pneumonia to adjust for the reduced time at risk in persons with a pneumonia (i.e., the person stops being at risk when diagnosed with a pneumonia) [26]. The time at risk was then multiplied by the number of participants in the study. We only considered cases with pneumonia and not episodes of pneumonia, i.e., each person was considered only once in the calculations. In a sensitivity analysis, we calculated a minimal follow-up time by using 0 days for the cases and a maximal follow-up time by using the full follow-up time for all patients.

### 2.6. Risk-of-Bias Assessment

Risks of bias were assessed for all included studies using a “Quality assessment checklist for prevalence studies” developed by Hoy et al. [27]. The tool consists of 10 questions, and each question can be answered as “yes”, defined as low risk of bias or “no”, defined as high risk of bias. The quality assessment questions addressed external validity (items 1–4) and internal validity (items 5–10) (Appendix A). We selected the five most important items for the aims of our study (items 2, 4, 6, 7, 10). We decided for a conservative procedure that the worst item out of these five items was decisive for the rating of the whole study. Uncertainties in rating of the risk of bias were resolved by discussion with two further authors (R.H., G.M.). The full risk-of-bias assessment is shown in Appendix A. We planned to perform sensitivity analyses comparing results from studies with a high risk of bias compared to studies with low risk of bias within the subgroups.

### 2.7. Data Synthesis

The incidence rates were used to calculate pooled incidence rates per subgroup (i.e., for each combination of the study characteristics) with a random-effects meta-analysis and were used to perform a meta-regression to provide the incidence-rate ratios for the independent variables: Setting, Level of Injury, Ventilation, Tracheostomy, and Dysphagia [28]. Five categorical variables were used to create the subgroups and as independent variables in the univariable and multivariable regression models: Setting (Acute, ICU, Rehabilitation, Post-Rehabilitation, Long-Term Ventilated, and Mixed), Level of Injury (studies with only persons with paraplegia, studies with only persons with tetraplegia, studies where more than 50% but less than 100% of the persons had paraplegia, studies where more than 50% but less than 100% of the persons had tetraplegia, and “mixed” for studies where the proportion of persons with paraplegia or tetraplegia was not reported), Ventilation (Not Ventilated or No Information, Ventilated, and Mixed), Tracheostomy (No, Yes, Mixed), Dysphagia (Not Mentioned, No, Yes). The categorical variable dysphagia was not entered in the multivariable meta-regression because of the low number (*n* = 2) of included studies with information on dysphagia.

Settings were divided into ICU, acute phase, rehabilitation phase, post-rehabilitation phase and mixed setting, in which the first three settings are linked to hospital-acquired pneumonia and post-rehabilitation is linked to community-acquired pneumonia. Mechanical ventilation means ventilation through a tracheostomy or an endotracheal tube [29]. Dysphagia is commonly diagnosed by a swallow evaluation at the bedside, a flexible fiberoptic endoscopy evaluation of swallowing, or a videofluoroscopic swallow study [30]. Dummy variables were created, and the category with the hypothesized lowest incidence was chosen as the reference category. The meta-regression was performed with a Poisson regression with a random intercept, which corresponds to a random effects meta-regression. Details on this method can be found here [31,32]. Statistical heterogeneity was evaluated using tau2 and the I^2^ statistic. Tau2 is between study variance of the incidence rates, and I2 describes the proportion of variation in incidence estimates that is due to genuine variation rather than sampling error. Values for I^2^ of 50% or greater were considered to show substantial heterogeneity [33,34]. Analyses were performed with Stata, version 17 and R (R Foundation for Statistical Computing, Vienna, Austria) and the packages meta [35] and metafor [36].

## 3. Results

### 3.1. Studies Identified

Literature searches identified 719 records, including duplicates. Figure 1 displays the flow of the inclusion of records. After exclusion through comparisons of titles and abstracts against inclusion criteria, 97 records were identified for detailed examination. In total, 71 records were excluded (Appendix A) with the reasons listed in Figure 1. The main reason was that in 54 studies the wrong patient population was examined (Figure 1). Mainly, data for the sub-analyses for the SCI population were missing and therefore the numbers for incidence of pneumonia could not be used for our analyses.

Finally, 24 records met inclusion criteria, and 19 studies could be included in the analyses for the incidence rates and incidence rate ratios. Of the 24 included studies, only 2 designed their study specifically to report on incidence of pneumonia [37,38]. The study sample sizes ranged from 14 to 18,693 (median = 90), and the studies were published between 2001 and 2020 and carried out in Europe (*n* = 7), Asia (*n* = 4) and the U.S. (*n* = 13). Of these 24 studies, 12 studies were prospective. Table 1 describes the study characteristics.

### 3.2. Risk of Bias

Appendix A is a tabular display of the overall risk-of-bias assessment. Of the 24 included papers, all studies were rated as having a high risk of bias; therefore, we did not undertake a risk-of-bias sensitivity analysis.

### 3.3. Incidence Rates per 100 Person-Days and Incidence-Rate Ratios

Meta-regression of 19 studies reporting on 34 study samples was conducted. Because Raab et al. and Smith et al. did not report cases per time units, both studies were excluded from the meta-regression. The study from McKinley et al. from 2002 and two studies from Shem et al. from 2011 and 2012 were also excluded because the use of patients in different studies is quite likely. Only the latest studies were included in the analyses to ensure that each patient is only included once in our analyses.

Figure 2 shows the incidence rates for each study and pooled for each subgroup. The incidence rates ranged from 0.06 to 3.98 per 100 person-days for the acute setting, from 0.27 to 7.21 per 100 person-days for the ICU setting, from 0 to 1.84 per 100 person-days for the rehabilitation setting, and from 0.03 to 0.96 for the post-rehabilitation setting. Figure 2 also shows that the heterogeneity remains high, even when subgroups are built with the combination of setting, level of injury, ventilation, tracheostomy, and dysphagia (high I2 in most subgroups).

Table 2 shows the univariable and multivariable incidence rate ratios. For the setting, the incidence-rate ratio was 8.20 (95% CI 2.21 to 30.39) for the ICU compared to the post-rehabilitation setting. The incidence-rate ratios for the other settings were not statistically significant. The only other significant incidence-rate ratios were for the ventilated and mixed versus the studies where no information was given for the mechanical ventilation.

In the multivariable meta-regression (adjusted for setting, level of injury, ventilation, and tracheostomy), the incidence rate was only significant for the mixed setting versus the post-rehabilitation setting (IRR 15.76, 95% Ci 1.31 to 189.45), and the studies with a mix of ventilated and non-ventilated patients versus those with no information on ventilation (IRR 5.07, 95% CI 1.58 to 16.25).

### 3.4. Sensitivity Analyses

The sensitivity analyses (Appendix A) with the two alternative calculations of the time under risk did not produce results that would change the conclusion.

## 4. Discussion

This systematic review and meta-regression of 24 studies analyzed the incidence of pneumonia in SCI. All studies had a high risk of bias with high heterogeneity, and this was evident even in the subgroup analyses. While pooled estimates of incidence would be useful to indicate the public health burden of pneumonia in SCI, we have only low confidence in our pooled estimates of the incidence, which ranged from 0.03 to 7.21 patients with pneumonia per 100 days. This low confidence results mainly because of (i) the design of the studies, which were not specifically designed to analyze the incidence of pneumonia, (ii) the reporting of the follow-up time (time at risk), (iii) the small sample sizes, (iv) the non-standardized reporting of the outcome variables and the risk factors (i.e., setting, level of injury, mechanical ventilation, tracheostomy, dysphagia), (v) not all studies had a longitudinal design, and (vi) the high risk of bias.

Despite this, our results suggest that shortly after the onset of a SCI, when the patient is in an ICU, the incidence of pneumonia was almost five times as high as in the time after subsequent discharge from the rehabilitation setting. Given that most pneumonia occurs early after the SCI (Figure 2), we propose the need for a greater focus on regular screening of pulmonary and respiratory muscle function in the ICU and implementation of potential strategies to enhance pulmonary and respiratory muscle function (e.g., physiotherapy and respiratory muscle training).

### 4.1. Overall Completeness of Evidence

The overall completeness of evidence with 24 identified studies appears to be sufficient to address the incidence of pneumonia in SCI, taking into account the wide 95% confidence intervals. Most studies reported the incidence of pneumonia in SCI, and a minority reported the point prevalence or the period prevalence of pneumonia. We therefore decided to focus on incidence estimates. The analyses of incidence rates had the limitation that the time under risk was not reported. For a correct follow-up time, the follow-up days should only be counted up to the diagnosis of a pneumonia, and most studies reported the overall follow-up time, i.e., including the days where a patient was already diagnosed with a pneumonia. Therefore, we had to estimate the follow-up time by adjusting the days for patients with pneumonia (i.e., we took only half of these days for cases).

### 4.2. Covartiates

#### 4.2.1. Setting

The clinical setting influences the incidence of pneumonia. Rates are considerably higher among patients hospitalized in ICUs compared with those in hospital wards [29,58]. These findings are confirmed in this systematic review; the incidence of pneumonia in SCI is also highest in ICU and decreases with time post injury (Figure 2). The clinical setting varied between the studies we examined. For example, in some studies pneumonia was identified at ICU during the acute phase of SCI, and in others pneumonia was identified in the rehabilitation phase or later in the outpatient setting.

#### 4.2.2. Level of Injury

We investigated the incidence of pneumonia according to the level of injury. We divided the level of injury into tetraplegia and paraplegia because the degree of respiratory impairment depends on the level of injury, with higher levels of injury causing greater impairment [15,59]. Generally, the incidence of pneumonia is higher in high-level tetraplegia in comparison to low-level tetraplegia and paraplegia [6]. However, one study in this review directly compared patients with tetraplegia and paraplegia with comparable personal/baseline characteristics [54] and could not confirm this statement.

#### 4.2.3. Mechanical Ventilation

For patients receiving mechanical ventilation, the risk of pneumonia is increased 3- to 10-fold [29,58,60]. The incidence of ventilator-associated pneumonia ranged from 8% to 28% [58,61]. Ventilator-associated pneumonia is defined as pneumonia occurring >48 h after endotracheal intubation [62]. Some use the term hospital-acquired pneumonia to denote any pneumonia developing in the hospital (including ventilator-associated pneumonia) and others exclude ventilator-associated pneumonia from the hospital-acquired pneumonia designation [62]. Therefore, the comparability of the literature is complicated by inconsistent usage of the terms. Four of the studies in this review used the term “ventilator-associated pneumonia” [37,42,43,55] and one study used the term “hospital-acquired infection” [63]. The difference between successful or unsuccessful weaning from mechanical ventilation showed differences in the incidences of pneumonia, but this relies on only one study and hence no conclusion can be drawn. From all pneumonia events in hospital, about 60% occur in non-ventilated patients with SCI [64]. The incidence of hospital-acquired pneumonia is low (1.6%) in the non-ventilated general population [65]. These 1.6% correspond to about 22% of all hospital-acquired infections.

#### 4.2.4. Tracheostomy

Tracheostomy seems likely to influence the incidence of pneumonia, but this relies only on one single study [39]. In this study, no statistical significance for the timing of tracheostomy and pneumonia was given. Other studies report that the rates of pneumonia in SCI cannot be reduced by the timing of tracheostomy [66,67]. However, the timing of the tracheostomy within 7 days of entry to ICU is associated with a shorter duration of mechanical ventilation and shorter length of ICU stay [66,67,68]. A review of patients with trauma in ICU summarized the impact of timing of tracheostomy on pneumonia, and it reported that some studies found a reduction in pneumonia but some did not [69].

#### 4.2.5. Dysphagia

Dysphagia seems to be an important variable linked to the incidence of pneumonia, but only a few studies provided data that could be used. The most common cause of death in patients with dysphagia due to neurological disorders is aspiration pneumonia [70,71]. This is defined as an infection caused by the inhalation of oropharyngeal secretions that are colonized by pathogenic bacteria [72]. Generally, 5% to 15% of cases with community-acquired pneumonia are aspiration pneumonia [72,73]. The included studies in this review did not formally consider aspiration pneumonia.

#### 4.2.6. General Aspects

Due to the different compositions of setting, level of injury, mechanical ventilation, tracheostomy, and dysphagia in our sub-analyses, a direct comparison with the general population is difficult.

Generally, to reduce pneumonia incidence, a rapid identification of infected patients and appropriate antimicrobial or other treatment is required [58]. Unfortunately, qualitative influence on incidence such as time since injury, type of bacterial pneumonia or smoking could not be included due to missing details in the studies. In the future, it would be valuable to have more thorough recording of study characteristics with a clear definition of pneumonia type, setting, and description of ventilation or dysphagia to facilitate future meta-analyses.

### 4.3. Strengths and Limitations

Our search strategy was planned to comprise extensive literature searches of several major electronic databases as well as contact with experts in the field. Despite these searches, we might miss eligible studies, in particular if they are not published in indexed peer-reviewed journals. This might lead to a reporting bias [74]. Our literature search revealed three papers that were unable to be translated, and this constitutes a potential selection bias. Generally, the included studies were conducted in high-income countries, and therefore, the incidence of pneumonia after SCI in low-income countries could not be reflected in this systematic review. Due to the specific patient group and the strictly defined research question, we did not expect a large number of studies for inclusion. The number of included studies in each single sub-analysis was low (Figure 2). We are aware that many of the included studies were interventional studies with multiple inclusion and exclusion criteria and therefore increased potential risk for pneumonia that could bias our outcomes as well, but in this way all types of pneumonia were included to get an overview of the full picture in SCI. For the covariate setting, the acute phase is not a standardized term and therefore the time period was defined differently in the included studies or was not defined at all; this can lead to a mixture of different times post injury. Even if we had already tried to specify potential sources of clinical diversity by defining strict inclusion criteria in the protocol, we still had high heterogeneity according to the statistical I2 test (Figure 2). Most studies reported cumulative incidences, i.e., the number of events divided by the number of participants. Incidence rates (cases with pneumonia divided by the observed person-time) would be a better statistic, but few studies reported the time the persons were at risk. A further important source of clinical heterogeneity can be the insufficient definition of diagnostic criteria for pneumonia in a number of the included studies (Appendix A). Usually, with high diversity in a systematic review, conclusions need to be interpreted with caution or seen as hypotheses. Finally, we relied on the quality and quantity of available published information. Nonetheless, to our knowledge, this is the first attempt at a systematic review of the incidence of pneumonia in SCI.

## 5. Conclusions

The main finding of this systematic review and meta-regression is that we found large heterogeneity in the reporting of the incidence of pneumonia, and we therefore should be cautious with interpreting the results. Our overall incidence ranged from 0.03 to 7.21 patients with pneumonia per 100 days, with higher incidence in the acute and ICU setting than later after injury. Large longitudinal studies with a standardized reporting of risk factors, pneumonia, and detailed time under observation are needed. Nevertheless, this review showed that pneumonia is still a clinically relevant complication, and special attention to pneumonia prevention should focus on the ICU setting and patients with complete tetraplegia. We need to focus more on regular screening of pulmonary and respiratory muscle function in the ICU and doing what we can to enhance function (e.g., by respiratory muscle training).

## Figures and Tables

**Figure 1 jcm-11-00211-f001:**
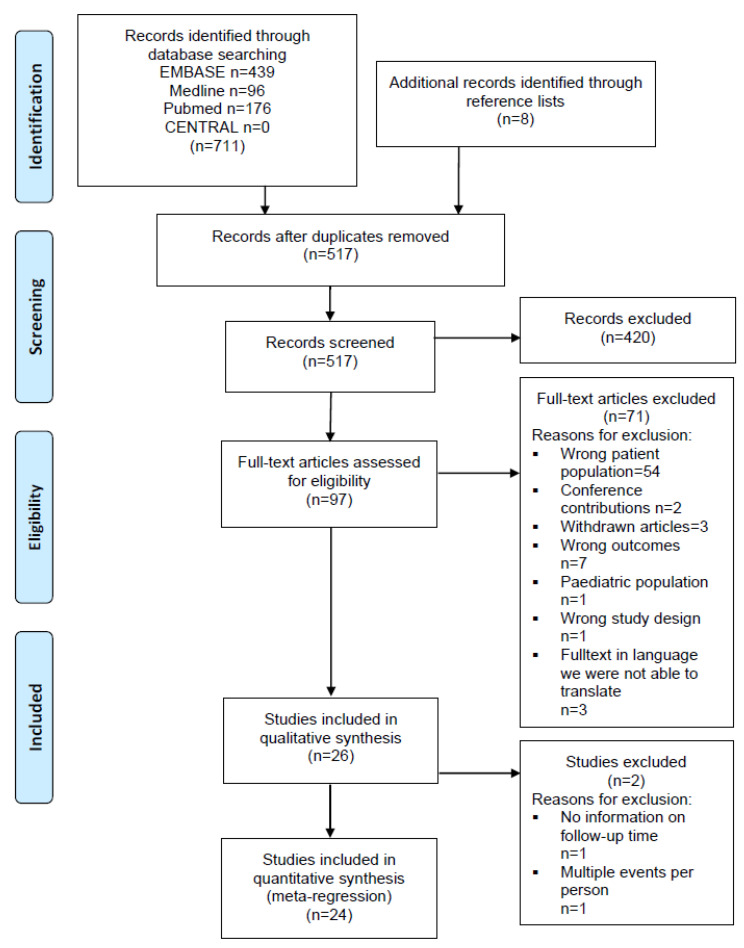
PRISMA flow diagram. Systematic review of incidence studies of pneumonia in persons with spinal cord injury.

**Figure 2 jcm-11-00211-f002:**
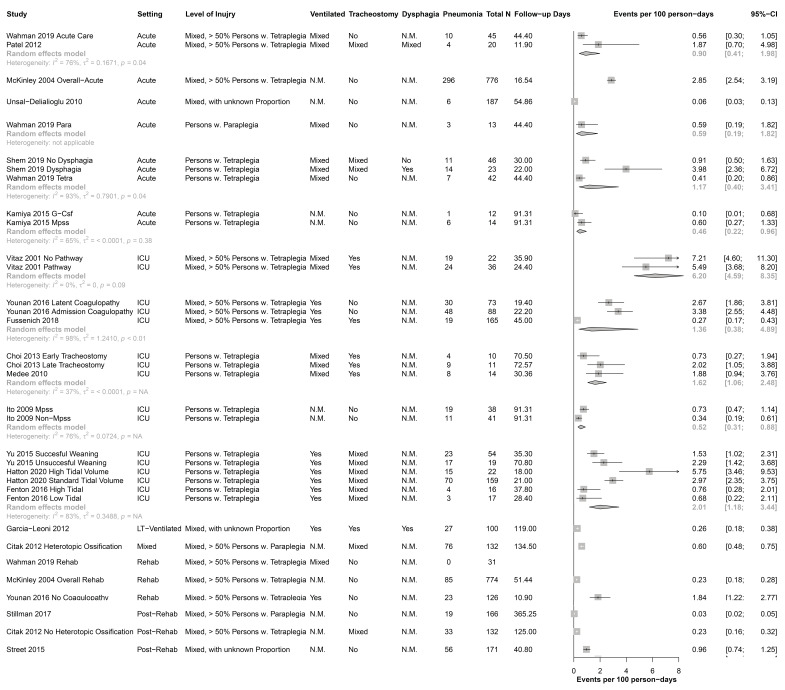
Forest Plot of all included study samples, without pooled results. CI = confidence interval; Fup, Follow up; G-Csf, granulocyte colonystimulating factor; ICU, Intensive Care Unit; LT-Ventilated, long-term ventilated; MPSS, high-dose methylprednisolone sodium succinate; n.m., not mentioned; Para, paraplegia; Rehab, rehabilitation; Tetra, tetraplegia; Ventilation, mechanical ventilation.

**Table 1 jcm-11-00211-t001:** Characteristics of 24 included studies with 34 subgroups.

Citation	Year of Publication	Country	Type of Study	Sample Size (*n*)	Setting	Mean Age (±SD)	Sex Male(%)	SCI Classification Using AIS (%)	SCI Level of Injury (%)	Mechanical Ventilator Depen-Dency (%)
Choi et al. [39] Early Tracheostomy	2013	Korea	retro-spective	10	ICU	54 ± 14	90	A 20 B 20 C 60 D 0	Tetra 100	70
Choi et al. [39] Late Tracheostomy	2013	Korea	retrospective	11	ICU	46 ± 17	91	A 55 B 9 C 27 D 9	Tetra 100	100
Citak et al. [40] Heterotopic Ossification	2012	Germany	pro-spective	132	Mixed	43 ± N.M.	84	A 83 B,C,D 17	Tetra 55 Para 45	N.M.
Citak et al. [40] No Heterotopic Ossification	2012	Germany	pro-spective	132	Mixed	49 ± N.M.	77	A 46 B,C,D 54	Tetra 49 Para 51	N.M.
Fenton et al. [41] High Tidal	2016	USA	pro-spective	16	ICU	39 ± 13	88	A,B,C 100	Tetra 100	100
Fenton et al. [41] Low Tidal	2016	USA	pro-spective	17	ICU	27 ± 7	65	A,B,C 100	Tetra 100	100
Fussenich et al. [42]	2018	Germany	retro-spective	165	ICU	57 ± 17	79	A 46 B 12 C 32 D 11	Tetra 82 Para 19	100
Garcia-Leoni et al. [37]	2010	Spain	pro-spective	100	LT-Ventilated	49 ± 17	75	N.M.	Tetra 58 (missing data N.M.)	100
Hatton et al. [43] HighTidal Volume	2020	USA	retro-spective	22	ICU	40 (27–51)	77	A 73 B,C,D 27	Tetra 100	100
Hatton et al. [43] Standard Tidal Voluma	2020	USA	retro-spective	159	ICU	53 (35–70)	79	A 38 B,C,D 62	Tetra 100	100
Ito et al. [44] Mpss	2009	Japan	pro-spective	38	ICU	55 ± N.M.	79	A 26 B 11 C 29 D 34	Tetra 100	N.M.
Ito et al. [44] Non-Mpss	2009	Japan	pro-spective	41	ICU		80	A 27 B 27 C 29 D 17	Tetra 100	N.M.
Kamiya et al. [45] G-Csf	2015	Japan	retro-spective	28	Acute	58 (38–72)	75	A 7 B 14 C 29 D 50	Tetra 100	N.M.
Kamiya et al. [45] Mpss	2015	Japan	retro-spective	34	Acute	61 (18–85)	79	A 26 B 9 C 32 D 32	Tetra 97 Unclear 3	N.M.
* McKinley et al. [46]	2004	USA	retro-spective	654	Acute	38 ± 16	79	A 49 B,C,D 51	Tetra 55 Para 45	N.M.
McKinley et al. [47] Non-Traumatic	2002	USA	pro-spective	38	Acute	55 ± 14	50	A 5 B 29 C 34 D 31	Tetra 34 Para 66	N.M.
McKinley et al. [47] Traumatic	2002	USA	pro-spective	79	Acute	39 ± 16	87	A 46 B 22 C 18 D 15	Tetra 42 Para 58	N.M.
Medee et al. [38]	2010	France	retro-spective	14	ICU	41 ± 18	57	A,B 100	Tetra 100	64
Patel et al. [48]	2012	USA	retro-spective	20	Acute	76 ± n.m	65	A 55 B 10 C 15 D 20	Tetra 50 Para 5 Central Cord 45	N.M.
Raab et al. [49]	2016	Switzer-land	retro-spective	307	Mixed	53 ± 15 53 ± 18	81	A 58 B 20 C 13 D 9	N.M.	N.M.
Shem et al. [19] No Dysphagia	2011	USA	pro-spective	17	Acute	35 ± 12	71	N.M.	Tetra 100	41
Shem et al. [19] Dysphagia	2011	USA	pro-spective	12	Acute	49 ± 21	83	N.M.	Tetra 100	67
Shem et al. [20] No Dysphagia	2012	USA	pro-spective	24	Acute	36 ± 13	71	A 54 B,C,D 46	Tetra 100	46
Shem et al. [20] Dysphagia	2012	USA	pro-spective	16	Acute	51 ± 18	88	A 25 B,C,D 75	Tetra 100	88
Shem et al. [30] No Dysphagia	2019	USA	pro-spective	53	Acute	39 ± 17	79	A 47 B,C,D 53	Tetra 100	40
Shem et al. [30] Dysphagia	2019	USA	pro-spective	23	Acute	48 ± 19	91	A 35 B,C,D 65	Tetra 100	65
Smith et al. [50]	2007	USA	retro-spective	18.693	Mixed	56 ± 14	98	A 24 B,C,D 27 Unknown 32	Tetra 33 Para 21 Missing 36	N.M.
Stillman et al. [51]	2017	USA	pro-spective	169	Post-Rehab	41 ± 16	79	A,B,C 48 D 50 Unknown 2	Tetra 23 Para 65 Unknown 2	N.M.
Street et al. [52]	2015	Canada	pro-spective	171	Post-Rehab	47 ± 20	81	N.M.	N.M.	N.M.
Unsal-Delialioglu et al. [53]	2010	Turkey	retro-spective	392	Acute	37 ± 14	76	A 52 B 11 C 19 D 18	N.M.	N.M.
# Wahman et al. [54] Para	2019	Sweden	pro-spective	45	Acute	55 ± 17	60	A 29 B,C,D 71	Tetra 71 Para 29	Yes (number N.M.)
Younan et al. [55] Latent Coagulopathy	2016	USA	retro-spective	73	ICU	39 ± 17	82	N.M.	° Tetra N.M. Para N.M.	100
Younan et al. [55] Admission Coagulopathy	2016	USA	retro-spective	88	ICU	39 ± 20	81	N.M.	° Tetra N.M. Para N.M.	100
Younan et al. [55] No Coagulopathy	2016	USA	retro-spective	126	Rehab	44 ± 16	82	N.M.	° Tetra N.M.Para N.M.	100
Yu et al. [56] Successful Weaning	2015	Taiwan	retro-spective	54	ICU	49 ± 19	83	∞ A 34B 10 C 10 D 4 Unknown 43	Tetra 100	100
Yu et al. [56] Unsuccessful Weaning	2015	Taiwan	retro-spective	19	ICU	64 ± 17	84	∞ A 34 B 10 C 10 D 4 Unknown 43	Tetra 100	100
Vitaz et al. [57] No Pathway	2001	USA	retro-spective	22	ICU	34 ± 10	N.M.	N.M.	Tetra 86 Para 14	Yes (number N.M.)
Vitaz et al. [57] Pathway	2001	USA	pro-spective	36	ICU	33 ± 15	N.M.	N.M.	Tetra 89 Para 11	Yes (number N.M.)

ASIA = American Spinal Injury Association; CI = Confidence Interval; Gcf = granulocyte colonystimulating factor; Mpss = methylprednisolone sodium succinate; N.M. = not mentioned in study; Para = Paraplegia; SCI = Spinal Cord Injury; Tetra = Tetraplegia. All numbers are rounded up or down to a full turn-out. Mechanical ventilation: N.M.—this means that in the study it is not mentioned whether the participants needed mechanical ventilation or not; YES (number N.M.)—this means that in the study it is mentioned that the participants were ventilated, but the exact number of ventilated participants is not given; NO—the participants in the study were not ventilated. * McKinley et al. (2004) [46] only reported overall values. For our pneumonia analyses, we used the subgroups Overall_Acute/Overall_Rehab. # Wahman et al. (2019) [54] only reported overall values. For our pneumonia analyses, we used the subgroups Tetra/Para. ° Younan et al. (2016) [55] reported numbers higher than 100% with no reason given, and therefore we wrote N.M. ∞ Yu et al. (2015) [56] did not present a subdivision for ASIA for the subgroups Successful weaning/Unsuccessful weaning.

**Table 2 jcm-11-00211-t002:** Univariable and multivariable Meta-Regression including Setting, Level of Injury, Ventilation, Tracheostomy, and Dysphagia.

	Univariable Meta-Regression			Multivariable Meta-Regression		
Variable	Incidence Rate Ratio	95% CI	*p*-Value	Incidence Rate Ratio	95% CI	*p*-Value
Setting						
Post-Rehab (Reference)	1.00			1.00		
Acute	3.25	0.81 to 12.97	0.095	0.65	0.13 to 3.35	0.605
ICU	8.20	2.21 to 30.39	0.002	2.27	0.32 to 15.93	0.410
Long-Term Ventilation	1.33	0.12 to 14.6	0.817	0.97	0.05 to 17.64	0.983
Mixed	3.08	0.28 to 33.3	0.355	15.76	1.31 to 189.45	0.030
Rehab	3.23	0.49 to 21.41	0.224	0.75	0.1 to 5.9	0.785
Level of Injury						
Persons w. Paraplegia (Reference)	1.00					
Mixed, >50% Persons w. Paraplegia	0.28	0.02 to 4.62	0.371	0.20	0.01 to 0.01	0.296
Mixed, >50% Persons w. Tetraplegia	2.54	0.22 to 30.11	0.459	3.64	0.41 to 0.41	0.244
Persons w. Tetraplegia	2.03	0.18 to 23.46	0.571	1.93	0.21 to 0.21	0.562
Mixed, with unknown Proportion	0.49	0.03 to 7.17	0.601	1.99	0.16 to 0.16	0.596
Ventilation						
Not mentioned (Reference)	1.00					
Mixed	4.70	1.89 to 11.72	0.001	5.07	1.58 to 16.25	0.006
Ventilated	4.34	1.76 to 10.71	0.001	2.03	0.61 to 6.73	0.247
Tracheostomy						
No (Reference)	1.00					
Mixed	2.55	0.99 to 6.57	0.053	1.17	0.42 to 3.21	0.763
Yes	2.52	0.85 to 7.48	0.096	0.41	0.1 to 1.69	0.217
Dysphagia						
No (Reference)	1.00			*		
Not mentioned	0.95	0.07 to 12.18	0.968			
Mixed	1.93	0.05 to 72.32	0.722			
Yes	1.13	0.05 to 24.19	0.938			

* Too few studies with information on dysphagia (only two studies reported on dysphagia), therefore, we did not include dysphagia in the multivariable model. Incidence Rate Ratio: exp(coefficient): how many times the incidence per 100 days is higher compared to the reference category, (in multivariable analysis: controlled for all other variables).

## Data Availability

Detailed information on study data and analysis are available upon request from the corresponding author.

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
