# Peer review of "Systematic Review of Incidence Studies of Pneumonia in Persons with Spinal Cord Injury"

_jcm, 2021, doi:10.3390/jcm11010211_

Round 1
Reviewer 1 Report
I think this article is straightforward and easy to understand for readers.
However, there are several unclarified issues.
1) What is the main purpose of this study? If you can clarify the burden of SCI patients with pneumonia, what does it mean? It's better to state more clearly in the paper.
2) You concluded that "In the multivariable meta-regression the incidence in the ICU was more than twice as high (incidence rate ratio 2.28 [0.32-15.93])," but I think this is an overstatement. This analysis hardly shows the significance. Though you insist after this sentence, "this result was not statistically significant," nobody believes that this result is significant. I suggest you change the statement.
3) You mentioned multiple times that this study has a problem of heterogeneity, but you did not write down the I2 or funnel plot anywhere in the paper. I'd like to see those data to understand the heterogeneity.
4) From my standpoint of view, pneumonia is usually stratified to community-acquired, hospital-acquired and ventilator-associated pneumonia. It should be clarified, and it's better to additionally analyze the data using this stratification.
Author Response
We thank the reviewer for the constructive comments. Your expert input provided great guidance in improving the manuscript.
In the listing below, we provide a full account of how we have addressed the reviewers’ comments. We have also provided a marked-up version of the manuscript file that indicates all changes made.
Reviewer #1:
I think this article is straightforward and easy to understand for readers.
However, there are several unclarified issues.
1) What is the main purpose of this study? If you can clarify the burden of SCI patients with pneumonia, what does it mean? It's better to state more clearly in the paper.
The main purpose of the study was to provide an objective synthesis of the evidence about the incidence of pneumonia in persons with SCI using five covariates: clinical setting, the level of injury, use of mechanical ventilation, presence of tracheostomy, and dysphagia. Selection of these five covariates are either based on our clinical experience but also on the current SCI literature. This purpose can be found in the last sentence of the introduction.
We further tried to clarify the burden of SCI patients with pneumonia and added the following sentences into the introduction:
“Respiratory complications are one of the main comorbidities after SCI, especially among persons with cervical and high thoracic injury. The underlying problem is paralysis of the respiratory muscles, which leads to poor mobilization of secretion, bacterial accumulation in the secretion and resultant respiratory infections. The higher the level of SCI, the greater the risk of respiratory complications.
2) You concluded that "In the multivariable meta-regression the incidence in the ICU was more than twice as high (incidence rate ratio 2.28 [0.32-15.93])," but I think this is an overstatement. This analysis hardly shows the significance. Though you insist after this sentence, "this result was not statistically significant," nobody believes that this result is significant. I suggest you change the statement.
We agree with the reviewer and tried to moderate the statement.
We exchanged the statement in the abstract and the discussion of the paper with:
“In the multivariable meta-regression the incidence rate ratios showed very wide CIs, which does not allow a clear conclusion concerning the risk of pneumonia in the different stages after a SCI.”
3) You mentioned multiple times that this study has a problem of heterogeneity, but you did not write down the I2 or funnel plot anywhere in the paper. I'd like to see those data to understand the heterogeneity.
The reviewer is absolutely right that the I2 are important numbers to estimate the heterogeneity in a systematic review.
In the results and discussion sections the following sentences clarifies that all I2 numbers can be found in Figure II (first row, below each subgroup):
“Figure 2 also shows that the heterogeneity remains high, even when subgroups are built with the combination of setting, level of injury, ventilation, tracheostomy and dysphagia (high I2 in most subgroups).”
“Even if we had already tried to specify potential sources of clinical diversity by defining strict inclusion criteria in the protocol, we still had high heterogeneity according to the statistical I2 test (Figure 2).”
4) From my standpoint of view, pneumonia is usually stratified to community-acquired, hospital-acquired and ventilator-associated pneumonia. It should be clarified, and it's better to additionally analyze the data using this stratification.
We agree with the reviewer and an analysis using the stratification into community-acquired, hospital-acquired and ventilator-associated pneumonia would be great.
Unfortunately, in most included studies this information about community-acquired, hospital-acquired and ventilator-associated pneumonia is not given. Furthermore, as we already wrote in the discussion-section, there is no standardized usage/definition of the terms ‘community-acquired’, ‘hospital-acquired’ and ‘ventilator-associated’ pneumonia, see below.
“Ventilator associated pneumonia is defined as pneumonia occurring >48 hours after endotracheal intubation. Some use the term hospital-acquired pneumonia to denote any pneumonia developing in the hospital (including ventilator associated pneumonia) and others exclude ventilator-associated pneumonia from the hospital acquired pneumonia designation. So, the comparability of the literature is complicated by inconsistent usage of the terms.”
Reviewer 2 Report
The systematic review by Raab et al., shows the incidence of pneumonia using available human SCI data. It is well written and easy to follow
Author Response
We thank the reviewer for this positive feedback.
Round 2
Reviewer 1 Report
Thank you for your correction. I think your logics become more robust.